# Expression of pyrethroid metabolizing P450 enzymes characterizes highly resistant *Anopheles* vector species targeted by successful deployment of PBO-treated bednets in Tanzania

Johnson Matowo[1]◉*, David Weetman[2]◉, Patricia Pignatelli[2], Alexandra Wright[3], Jacques D. Charlwood[3], Robert Kaaya[1], Boniface Shirima[1], Oliva Moshi[1], Eliud Lukole[4], Jacklin Mosha[4], Alphaxard Manjurano[4], Franklin Mosha[1], Mark Rowland[3], Natacha Protopopoff[3]

1 Department of Medical Parasitology and Entomology, Kilimanjaro Christian Medical University College, Moshi, Tanzania, 2 Department of Vector Biology, Liverpool School of Tropical Medicine, Liverpool, United Kingdom, 3 Department of Disease Control, London School of Hygiene and Tropical Medicine, London, United Kingdom, 4 Department of Parasitology, National Institute for Medical Research, Mwanza Medical Research Centre, Mwanza, Tanzania

◉ These authors contributed equally to this work.
* johntowo@yahoo.com

**Data Availability Statement:** The microarray data generated are deposited in Array Express with

## Abstract

Long lasting insecticidal nets (LLINs) are a proven tool to reduce malaria transmission, but in Africa efficacy is being reduced by pyrethroid resistance in the major vectors. A previous study that was conducted in Muleba district, Tanzania indicated possible involvement of cytochrome P450 monooxygenases in a pyrethroid resistance in *An. gambiae* population where pre-exposure to piperonyl butoxide (PBO) followed by permethrin exposure in CDC bottle bioassays led to partial restoration of susceptibility. PBO is a synergist that can block pyrethroid-metabolizing enzymes in a mosquito. Insecticide resistance profiles and underlying mechanisms were investigated in *Anopheles gambiae* and *An. funestus* from Muleba during a cluster randomized trial. Diagnostic dose bioassays using permethrin, together with intensity assays, suggest pyrethroid resistance that is both strong and very common, but not extreme. Transcriptomic analysis found multiple P450 genes over expressed including CYP6M2, CYP6Z3, CYP6P3, CYP6P4, CYP6AA1 and CYP9K1 in *An. gambiae* and CYP6N1, CYP6M7, CYP6M1 and CYP6Z1 in *An. funestus*. Indeed, very similar suites of P450 enzymes commonly associated with resistant populations elsewhere in Africa were detected as over expressed suggesting a convergence of mechanisms across Sub-Saharan African malaria vectors. The findings give insight into factors that may correlate with pyrethroid PBO LLIN success, broadly supporting model predictions, but revision to guidelines previously issued by the World Health Organization is warranted.

accession numbers E-MTAB-10579 and E-MTAB-10580. Microarray analysis results and quantitative PCR data and results are provided within the supplementary materials.

**Funding:** The study was supported by the Welcome Trust through Malaria Capacity Development Consortium (MCDC), a Post Doctoral award to JM (grant No. ITDCBC9410) with additional support from the Joint Global Health Trials Scheme of the UK, Department for International Development, Medical Research Council, and Welcome Trust (MR/L004437/). The funders had no role in study design, data collection and analysis, decision to publish, or preparation of the manuscript.

**Competing interests:** The authors have declared that no competing interests exist.

## Introduction

The massive scale-up of insecticide treated bednets (ITNs) and in particular long-lasting insecticide-treated nets (LLINs) across sub-Saharan Africa has been the predominant factor in reducing malaria morbidity and deaths since the turn of the century [1]. Unfortunately, the number of malaria cases rose in several African countries in 2016 and 2017, and more widespread resurgence is possible [2]. Funding constraints in the most endemic countries is one factor holding back recent progress [3]. Although less easy to quantify, another key factor in this resurgence is resistance among the vectors to pyrethroids (used for all LLIN treatments [4]), which is now widespread. Although less common than pyrethroid resistance, resistance to other insecticide classes used in vector control is emerging in many regions of sub-Saharan Africa [5–8]. To combat resistance to insecticides, and to pyrethroids in particular, the WHO has developed *The Global Plan for Insecticide Resistance Management* (GPIRM) [9]. Despite the prevalence of strong pyrethroid-resistance, many malaria-endemic countries have yet to align their vector control strategies to those of the GPIRM, in part because of a continued dependence on pyrethroid-treated LLINs [5]. The advent of next generation LLINs—not solely treated with pyrethroids—has been urgently awaited. The first of these bi-treated nets combines a pyrethroid with a non-insecticidal synergist piperonyl butoxide (PBO-LLIN). The aim is to improve pyrethroid efficacy, primarily by inhibiting enzymes involved in insecticide detoxification processes [10].

The main mechanisms of pyrethroid resistance in *An. gambiae* involve mutations to the voltage-gated sodium channel (*Vgsc*) target-site and metabolic resistance [11].

In *An. gambiae*, over expression of a handful of P450 genes from the CYP6 and CYP9 subfamilies have been repeatedly associated with pyrethroid resistance in field populations of *An. gambiae* in West and West-Central Africa [11]. When multiple mutations combine, they may lead to high-levels of resistance and are likely to seriously threaten the efficacy of malaria control programs [12]. Fewer transcriptomic studies have been performed in East Africa and it is unclear whether a similar concentration of metabolic resistance on a few P450 genes occurs. Although resistance-conferring *Vgsc* mutations are absent in *An. funestus*, metabolic resistance alone seems capable of producing high levels of resistance, which has been associated with control failure [13]. As with *An. gambiae*, a limited suite of CYP6 and CYP9 pyrethroid-metabolizing cytochrome P450s are involved in pyrethroid resistance in *An. funestus*, although the relative importance of specific genes varies geographically [14]. Widespread dependence of high-level resistance on pyrethroid metabolizing P450 enzymes in the major malaria vectors is promising for the efficacy of PBO-LLINs.

A recent four-armed cluster randomized trial (RCT) conducted in Muleba district, in Tanzania [15] which compared the effect of a standard pyrethroid LLIN with PBO-LLINs, each with or without Indoor residual spraying (IRS). Study arms with IRS (using the organophosphate pirimiphos-methyl Actellic) or PBO-LLIN had a strong reduction in malaria transmission compared to the LLINs alone arm. Thus, the implementation of either PBO or the different insecticide class for IRS overcame pyrethroid resistance [15]. In Uganda, a recently published second trial supports the recommendations given to PBO-LLIN [16].

Following results from the Muleba trial the WHO issued a policy statement that the deployment of PBO-LLIN should be considered in areas where the main malaria vector(s) have pyrethroid resistance. This is when resistance is (a) confirmed (b) of intermediate level (10–80% mortality) in diagnostic dose bioassays, and (c) at least partially conferred by a (P450) mono-oxygenase-based resistance mechanism [17]. With higher costs and also limitations to supplies, the question of when and where to deploy PBO- LLINs at an operational scale for malaria control is a crucial consideration for international agencies and national malaria control programmes. Prior to the trial, *An. gambiae* was the predominant malaria vector in

Muleba district with a high frequency of resistance to pyrethroids in diagnostic dose bioassays, attributed at least in part to near-fixation of the *Vgsc*1014S mutation [18]. Pyrethroid susceptibility in bioassays increased significantly if female mosquitoes were exposed to PBO indicating likely involvement of metabolic resistance mechanisms as well as the *Vgsc* mutation also referred to as *kdr* mutation. [19]. However, in Muleba, and in Tanzania generally, studies of specific genes involved in resistance have been limited to *Vgsc* mutations in *An. gambiae* [20] and have not been undertaken with *An. funestus*.

The present study, was aimed at characterizing molecular and metabolic resistance mechanisms present in intensely pyrethroid resistant populations of *An. gambiae* and *An. funestus*, and sought to describe the phenotypic and genetic insecticide resistance profiles of malaria vectors in Muleba. We hypothesize that the frequency of pyrethroid resistance in Muleba populations of *An. gambiae s.s* and *An. funestus* is high, accompanied by over expression of key candidate P450s genes.

## Materials and methods

### Consent and ethical clearance

Written consent was obtained from household heads before collecting mosquitoes inside the houses. This study was part of a cluster randomized trial that was conducted according to the Declaration of Helsinki and the International Guidelines for Ethical Review of Epidemiological Studies. Ethical clearance was obtained from the Medical Research Coordinating Committee (MRCC) of the National Institute for Medical Research (NIMR), London School of Hygiene and Tropical Medicine (LSHTM) and Kilimanjaro Christian Medical University College (KCMUCo).

### Study area and mosquito collections

The study was conducted in Muleba district, on the western shore of Lake Victoria in Tanzania. The area is characterized by high malaria prevalence and the presence of the two major malaria vectors *An. gambiae s.s.* and *An. funestus* [15].

It was the site for a RCT that involved 40 villages each of which received one of the four possible treatments in February 2015; standard LLIN, PBO-LLIN, IRS and LLIN, IRS and PBO-LLIN. Olyset net, the standard LLIN (Sumitomo Chemicals, Japan) was treated with 2% permethrin while the PBO-LLIN, Olyset Plus (Sumitomo Chemicals, Japan) contained 2% permethrin and 1% PBO. For IRS, Actellic 300CS (Syngenta, Switzerland) a microencapsulated pirimiphos-methyl was applied at a dosage of 1 g/m$^2$ on the inside walls and ceiling of each house. Detailed information about the study site and interventions is published elsewhere [15].

One or two villages with high *Anopheles* density were selected, in each of the four RCT study arms (Fig 1). Kakoma received standard LLIN, Kishuro PBO-LLIN, Bweyenza received IRS and LLIN and the remaining two villages Kiteme & Kyamyorwa each received IRS and PBO-LLIN. *Anopheles* were sampled between November 2014 and January 2015 in all the villages at baseline, before intervention deployment in February 2015. Due to the reduction in mosquitoes observed in villages receiving PBO-LLIN and/or IRS, collections were discontinued and only Kakoma remained as a sentinel site for resistance throughout the trial. It was sampled in May and June 2016. Another village allocated to the LLIN arm, Kabirizi was selected in 2017 due to the high number of *An. funestus* found there.

### Mosquito collection and identification

**Wild mosquitoes.**   Indoor resting *Anopheles* mosquitoes were collected from different houses in the villages between 0600 and 0830 using mouth and/or prokopack aspirators. Blood fed female *An. gambiae* and *An. funestus* were kept for three days for blood digestion and used

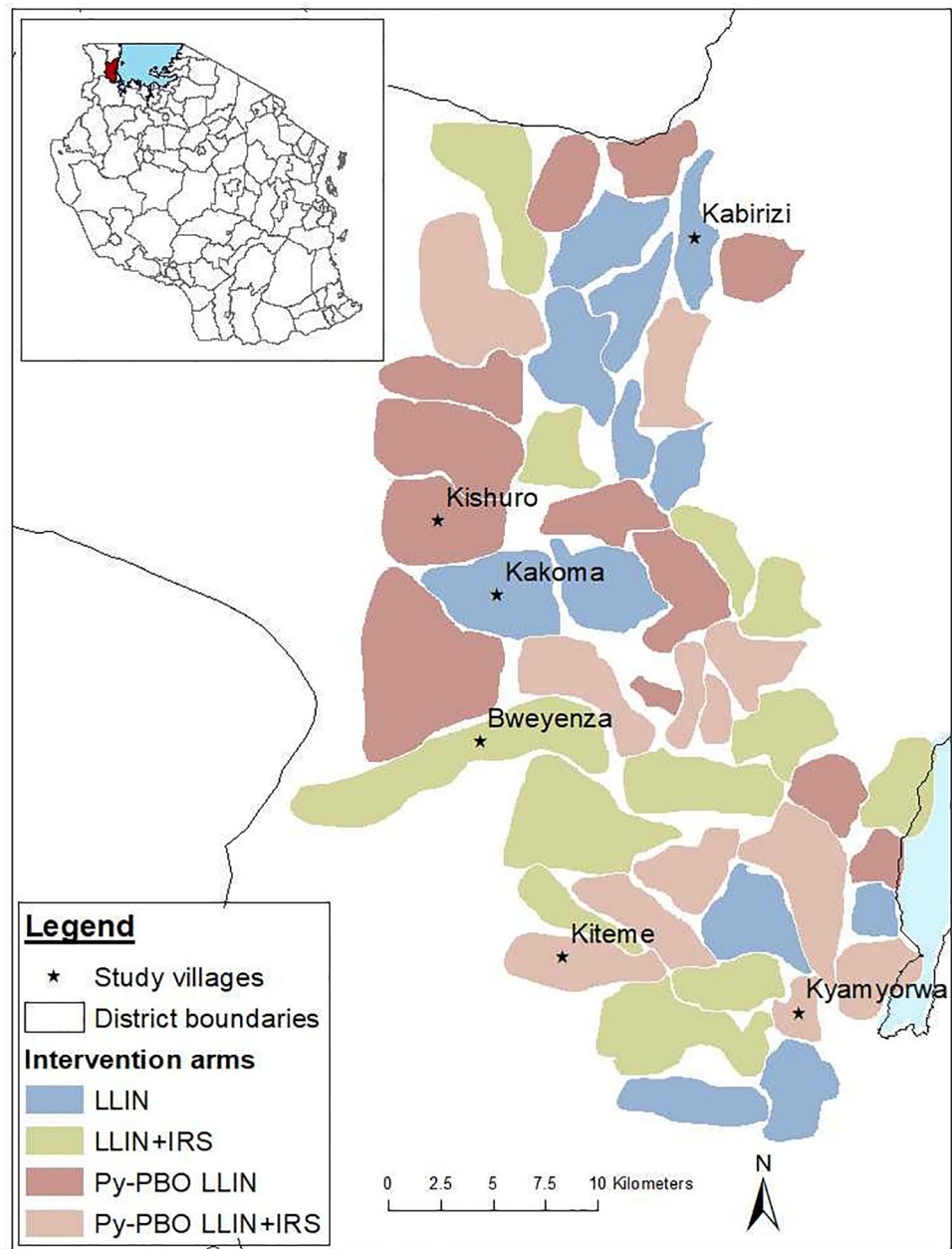

**Fig 1. Map of Tanzania showing the location of Muleba district and the study villages.**

for all WHO diagnostic dose susceptibility and CDC synergist and intensity bottle assays. F1 progeny used for target site mutations and metabolic gene expression characterization were reared, in the PAMVERC accredited laboratory in Moshi, from adult *An. gambiae* s.s. collected from Kakoma and *An. funestus* from Kabirizi in 2017. Morphological identification of the field caught females *Anopheles* was done according to the key of Gillies and Coetzee [21].

The SINE-PCR method of Santolamazza et al. [22] was used to discriminate members of *An. gambiae* species complex tested during microarray, while TaqMan assays [23] were performed on a sub-sample exposed to CDC bottle and WHO bioassays. Member species of the *An. funestus* group were identified using a cocktail PCR described by Koekemoer et al. [24].

**Laboratory susceptible strains.**   Three susceptible laboratory strains, namely *An. gambiae s.s* (Kisumu), *An. coluzzii* (Ngousso) and *An. funestus* (FANG) reared at the Liverpool School of Tropical Medicine were used for microarray and qPCR and used as reference strains for gene expression assays. The FANG strain that originated from Southern Angola, colonized at Liverpool since 2002 was used as a reference strain for *An. funestus* gene expression. For *An. gambiae s.s* gene expression, we used both Kisumu and Ngousso as reference strains to make analysis more stringent since the Kisumu strain has been colonized in laboratory for so long that it may not resemble a wild population very well.

## Resistance assay with diagnostic concentration

To assess resistance status of malaria vectors in the study area (S1 Fig), wild caught female *An. gambiae* s.l. and *An. funestus* were exposed for one hour in WHO cylinders to papers procured from the WHO recommended supplier Universiti Sains Malaysia treated with diagnostic concentrations of either permethrin (0.75%) (pyrethroid), lambda-cyhalothrin (0.05%) (pyrethroid), bendiocarb (0.1%) (carbamate) or pirimiphos-methyl (0.25%) (organophosphate) [25]. These insecticides are currently used for LLIN and IRS treatment.

Mosquitoes were then transferred to a holding tube, provided with 10% sugar solution and their mortality was recorded 24 hours later. Approximately 100 mosquitoes (25 per replicate) were used per test.

Tests with control mortality exceeding 5% were excluded. This assessment was carried out to ascertain the frequency of insecticide resistance for each insecticide in Muleba populations of *An. gambiae s.s* and *An. funestus*.

## Synergist bioassays

Synergist assays with piperonyl butoxide (PBO) were undertaken in 2017 to identify the potential role of elevated mixed-function oxidases in resistance in *An. gambiae* s.l. (Kakoma) and *An. funestus* (Kabirizi). According to treatment mosquitoes were pre-exposed for one hour, either to 50 μg/ml PBO treated or acetone coated bottles, and transferred for 30 minutes into bottles treated with 21.5μg/ml permethrin or acetone. Mortality was recorded 24 hours post-exposure [26].

## Resistance intensity dose response assay

To establish the intensity of resistance, a dose response bioassay using modified Centers for Disease Control and prevention (CDC) bottle bioassays, [25, 26] were performed on female *An. gambiae* and *An. funestus* collected from Kakoma and Kabirizi villages in April-May 2016. Wheaton bottles were coated with concentrations of permethrin that gave between 5% and 95% mortality (5 μg/ml to 860 μg/ml for *An. gambiae s.l.*, 21.5 μg/ml to 215 μg/ml for *An. funestus* and 1.6 μg/ml to 21.5 μg/ml for susceptible *An. gambiae* Kisumu strain). Approximately 12 mosquitoes were aspirated into each bottle and knock-down recorded at the start and after 15- and 30-minutes exposure.

Mosquitoes were transferred to paper cups, provided with 10% sugar solution, and mortality recorded after 24 hours. Five to eight replicates were performed for each concentration alongside with a control bottle (coated with acetone).

## Genotyping of *kdr* and GSTe2 mutations

We genotyped *kdr* and GSTe2 mutations in F1 progeny from field-collected *An. gambiae s.s.* and *An. funestus*, respectively; each of these mutations have proven causative links with pyrethroid resistance [27, 28]. To genotype the nucleotide variants leading to *kdr* mutations in the

*An. gambiae* VGSC (L1014F or S), hydrolysis probe assays were undertaken as described by Bass et al. (2007) [28]. These used TaqMan primers and minor groove binding (MGB) probes (Applied Biosystems, UK) and SensiMix DNA kit (Quantace).

The qPCR was run on an MxPRO3005 thermal cycler and analysed from endpoint scatter plots using MxPRO software (Aligent technologies, Stratagene, USA). The qPCR cycling conditions for both L1014F and L1014S were 95˚C for 10 minutes, followed by 45 cycles of 95˚C for 15s and 63˚C for 45s.

A further TaqMan assay was used to assess the presence and role of the L119F-GSTe2 mutation, previously associated with DDT and pyrethroid resistance in *An. funestus* [29]. Reactions were run and analyzed as above with the following conditions. The final volume contained 1× SensiMix (Bioline, London, UK), 800 nM of each primer and 200nM of each probe and the PCR cycling conditions included an initial denaturation at 95˚C for 10 minutes, followed by 40 cycles of 95˚C for 10s and 60˚C for 45s.

## Transcriptome analyses

Transcriptome analyses were carried out to identify genes that were putatively involved in observed insecticide resistance in Muleba populations of An. *gambiae s.s.* and *An. funestus*. Both *An. gambiae s.s.* and *An. funestus* were collected in May-June 2017 from Kakoma and Kabirizi villages respectively.

F1 female and male *An. gambiae s.s.* and *An. funestus* were separated on the day of emergence and the males were discarded. Females were fed on 10% glucose solution until they were three days old. The mosquitoes were then killed instantly in ethanol and preserved in RNAlater, stored overnight at 4˚C then transferred to -20˚C for longer-term storage.

Fully-interwoven loop designs were used to compare transcriptome expression profiles of wild pyrethroid-resistant *An. funestus* from Kabirizi village to the insecticide susceptible *An. funestus* laboratory strain FANG. Wild, pyrethroid-resistant, *An. gambiae* from Kakoma village were compared to the insecticide susceptible laboratory strains Kisumu and Ngousso (S2 Fig). Each comparison in the two experiments consisted of four independent biological replicates of RNA from pools of ten females in a balanced design to mitigate any dye bias.

RNA was extracted from the four batches of ten female mosquitoes using the RNAqueous kit (Thermo Fisher) according to the manufacturer's instructions and treated with DNase I (Qiagen). Quality and quantity of the RNA were checked using a Nanodrop spectrophotometer (Nanodrop Technologies, Wilmington, DE, USA) and a Bioanalyser 2100 (Agilent Technologies, USA). Each extracted pool of RNA was labeled separately with cy3 and cy5 dyes using the Low Input Quick Amp Labelling Kit (Agilent Technologies, USA) according to the manufacturer's instructions.

Labeled samples were hybridized to a 60k-probe microarray for *An. funestus* (Agilent; A-MEXP-2374) [29], or, for *An. gambiae*, a 15k-probe microarray (Agilent; A-MEXP-2196 [14], using the Agilent gene expression hybridization kit (Agilent Technologies, USA). Slides were washed according to the manufacturer's instructions and scanned on an Agilent G2565CA microarray scanner. Data was extracted using Feature Extraction 12.0 software (Agilent Technologies, USA).

## Candidate gene expression analysis

Quantitative PCR analysis was performed on genes identified as candidates from the microarray experiments. Primers were designed using the NCBI primer BLAST [30]. Complementary DNA (cDNA) was synthesized from the same RNA samples used in the microarray experiments using oligo(dT)20 and SuperScript III (Invitrogen) according to the manufacturers'

instructions and purified through a DNA-binding column (Qiagen). The quality and quantity of cDNA was measured using a Nanodrop spectrophotometer (Nanodrop Technologies, Wilmington, DE, USA). To check the dissociation curve and estimate efficiencies, primer pairs were tested using one pool from the wild samples and one from a laboratory strain, for each species respectively, in a dilution series starting from approximately 20ng/μg of cDNA.

Primer pairs exhibiting a linear relationship between threshold cycle (Ct) values and template concentration in standard curves, and high PCR amplification efficiency were chosen for further analysis. The qPCR reactions were performed using the Agilent MXPro Real-Time PCR detection system (Agilent Technologies, Stratagene, USA). A total volume of 20μl contained 10μl Brilliant III SYBR Green, 0.6ul of each primer (at 10nM) 300 nM of primers, 1μl of cDNA (at 2ng/ul) and 7.8 μl sterile-distilled water. The thermal profile was as follows: 1 cycle 95°C for 3 min, 40 cycles of 95°C for 10s then 60°C for 10s. Four biological replicates were run for each sample, with three technical replicates. Two endogenous normalizing genes, ribosomal S7 and elongation factor tau, were amplified for each sample to control for variation in cDNA quantity (primer pairs used for each species are shown in S1 and S2 Tables).

### Data analysis

Diagnostic dose bioassay results were interpreted according to WHO criteria: A 24 hours mortality superior to 98% indicates susceptibility, mortality of 90 to 97% suspected resistance, and mortality of less than 90% confirmed resistance [25]. To assess resistance intensity, Diagnostic concentrations which killed 50% (LC$_{50}$) of wild *An. gambiae* and *An. funestus* were estimated by Probit analysis using Polo plus (version 1.0, Le Ora Software LLC). The LC$_{50}$ values were used to calculate resistance ratios between each wild population and the *An. gambiae* Kisumu susceptible reference strain for assessing resistance intensity. The *An. gambiae* Kisumu susceptible reference strain was used as a comparator for both species, owing to unavailability of a susceptible *An. funestus* strain in the testing laboratory in Muleba. A resistance ratio of 2 indicates potential resistance or suggestive of probable resistance while resistance ratio greater than 2 indicates resistance.

Loess normalization of microarray data was performed by LIMMA 2.4.1 [31], with analysis of expression using the MAANOVA package [32], with data processing using custom R scripts [14]. Two of the *An. funestus* arrays were excluded owing to damage to a slide, but the fully interwoven loop design used still provided robust results under such circumstances [33].

Statistical significance of probes was determined using strict criteria based on a multiple testing corrected probability (q-value) threshold (q<0.0001), effect size (fold change, FC >2 or FC <-2) and, for *An. gambiae*, cross-experiment replication criteria vs. both susceptible colonies. Gene expression for each target gene, was normalized using that of the endogenous genes and was then analysed relative to the susceptible strains of *An. gambiae* or *An. coluzzii* (Kisumu and Ngousso) or *An. funestus* (FANG) using the $2^{-\Delta\Delta Ct}$ method, correcting for PCR efficiency variation [34]. Differences in expression were tested in Microsoft Excel by comparing the delta Ct values between strains using either a Student's t-test or, if an F-test indicated significant heterogeneity of variances between groups, a Welch's t-test (t-test not assuming equal variances). For both F-tests and t-tests, a threshold of $p < 0.05$ was to used to assign a significant difference between treatments.

## Results

### Resistance assay with diagnostic concentration

In 2014, the number of mosquitoes exposed per insecticide and location varied between 89 and 171. The mortality rate of *An. gambiae* exposed to permethrin and lambda-cyhalothrin in WHO cylinder tests was low, with only one of the ten bioassays recording mortality above 10%

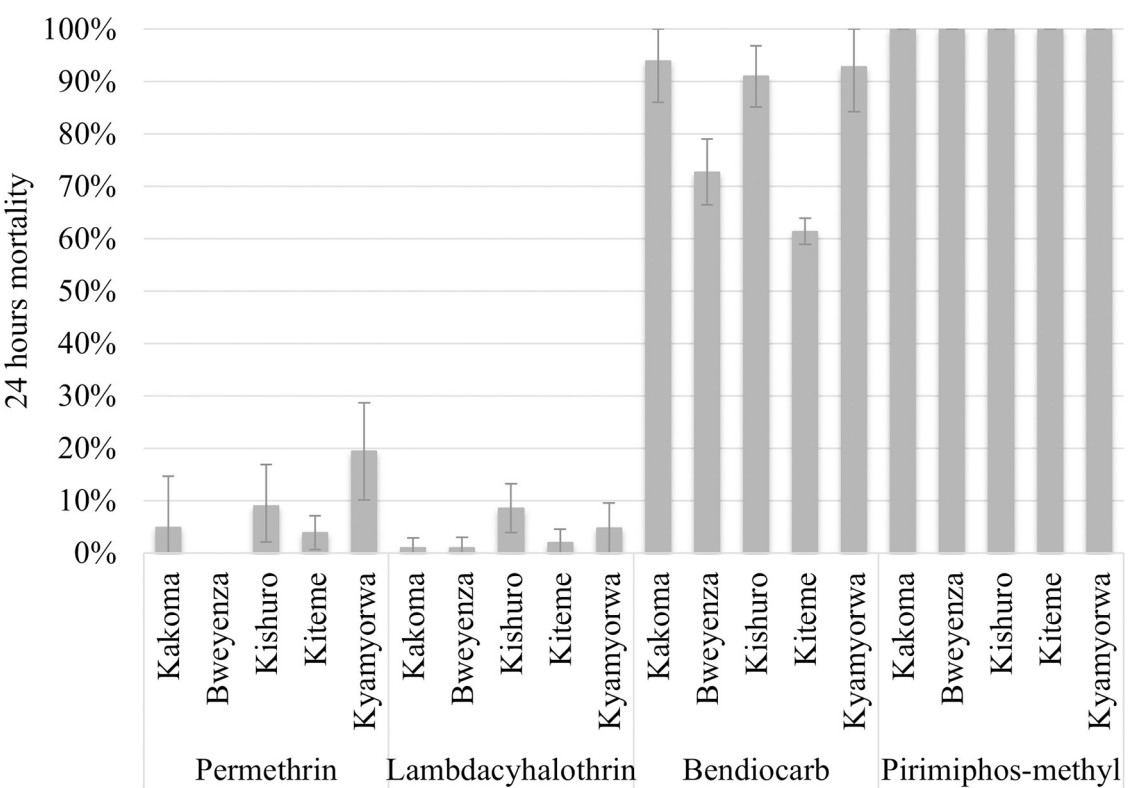

**Fig 2. Mortality of *Anopheles gambiae s.l* field populations exposed to permethrin, lambdacyhalothrin, bendiocarb and pirimiphos-methyl insecticides from five villages in 2014.**

(Fig 2). Mortality to bendiocarb, the active ingredient used previously for IRS, was much more variable ranging from around 90% in three villages to <35% in Kikagate village. All *An. gambiae* were fully susceptible to pirimiphos methyl, the active ingredient used for IRS in the trial. The number of *An. funestus* collected were too few to test at this time.

Mortality in *An. gambiae* s.l. collected from Kakoma in 2016, was 4% (95% CI: 0.8–7.9) similar to the baseline measurement for permethrin in 2014. Mortality in bendiocarb assays decreased from 94 to 45% ($\chi^2_2 = 54.3$, p<0.001) between 2014 and 2016, despite no operational exposure and despite the data coming from a mixture of two members of the *An. gambiae* complex. Pyrethroid and carbamate resistance was also observed in *An. funestus* from Kabirizi with 31% (95%CI: 24.9–37.1) mortality after exposure to permethrin and 80% (95%CI: 76.7–83.3) with bendiocarb.

In 2014, *An. gambiae* complex in Kakoma was composed of 93.5% (29/31) of *An.gambiae s. s.* (hereinafter called *An. gambiae*) in but in 2017 55.4% (46/79) with the remainder being *An. arabiensis*.

## Synergist bioassays

Due to strong resistance observed against permethrin, PBO synergist bioassay test using the CDC bottle assay were conducted in the same villages in 2017. After exposure to permethrin, *An. gambiae s.l.* mortality was 1.4% (95%CI: 0.1–3.6%) and increased to 18.1% (95%CI: 9.5–26.7%) when pre-exposed to PBO and 6.5% (95%CI: 0–15.9%) which increased to 53.2% (95% CI: 26.0–80.4) in *An. funestus*. Mortality in the PBO-only bottle (of 50 μg) was 2.8% and 17.3% for *An. gambiae* s.l. and *An. funestus* respectively and mortality in the control less than 2%.

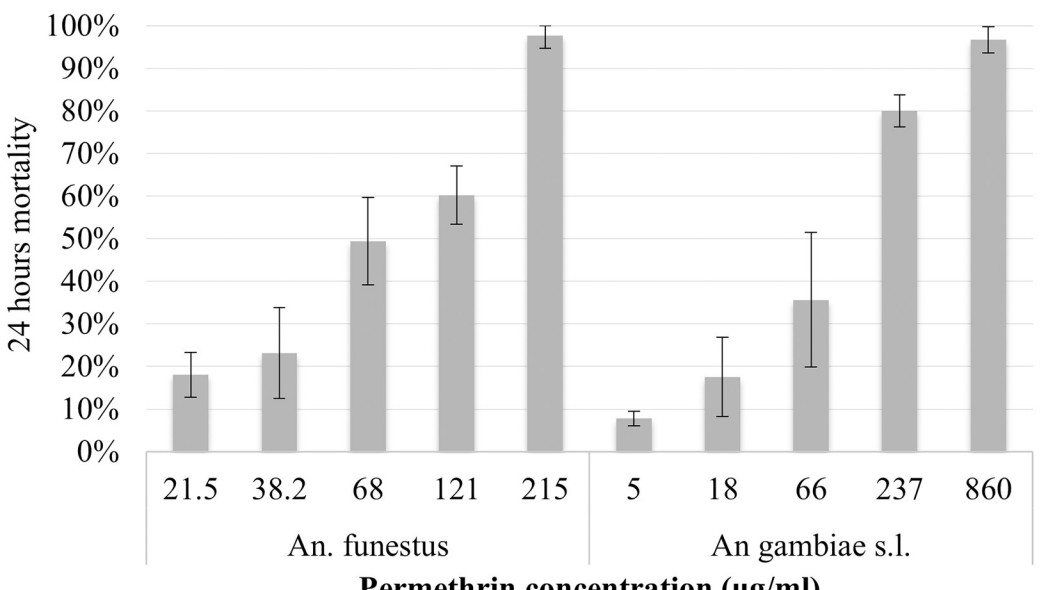

**Fig 3. 24 hours mortality in *An. gambiae s.l.* and *An. funestus* field population after exposure to different concentration of permethrin in intensity CDC bottle bioassays.**

## Resistance intensity dose response assay

In the 2016 collections done in Kakoma, a dose that was 40 times (860 μg/ml) the diagnostic dose of permethrin was required to produce a 24-hour mortality of 96.7% (88/91) in CDC bottle bioassays in *An. gambiae* s.l., and a dose that was 10 times the diagnostic concentration (215 μg/ml) was required to produce a mortality of 97.8% (87/89) in *An. funestus* (Fig 3). The *Anopheles* diagnostic dose in the CDC bottle bioassay for permethrin is 21.5 μg/ml.

*Anopheles gambiae*, *An. arabiensis* and *An. funestus* showed resistance ratios to permethrin of 63.2, 20.7 and 38.9, respectively compared to the susceptible *An. gambiae* colony (Kisumu). Although confidence intervals overlapped, this suggests a difference in mechanism or frequency of resistance between species, (Table 1).

## Molecular mechanisms

For *An. gambiae*, the 1014S *kdr* mutation was fixed (N = 227), whilst the 1014F mutation was rare, with only one homozygote detected with fewer than 10% F/S heterozygotes. No *kdr*

**Table 1. Lethal concentrations (LC$_{50}$ in μg/ml/bottle) and resistance ratios (RR$_{50}$) of permethrin against *An. gambiae s.l.* and *An. funestus* field population in intensity CDC bottle bioassays.**

| Species | Total exposed | LC$_{50}$* (95%CI**) | RR$_{50}$*** (95%CI) |
|---|---|---|---|
| Kisumu | 233 | 1.8 (1.1–2.4) | ref |
| *An. funestus* | 433 | 68.6 (59.2–79.4) | 38.9 (30.1–50.1) |
| *An. gambiae* | 195 | 111.5 (75.1–170.2) | 63.2 (42.0–94.9) |
| *An. arabiensis* | 51 | 36.6 (13.2–83.3) | 20.7 (11.2–38.3) |

*LC$_{50}$ = lethal concentration required to kill 50% of the population

**CI = Confidence interval

***RR$_{50}$ = resistance ratio or measure of resistance in an insect population, calculated by dividing the LC$_{50}$ of the resistant population by the LC$_{50}$ of the susceptible population.

mutations were detected in *An. arabiensis* (N = 245) [16]. The mutation L119F-GSTe2 potentially linked to resistance was not detected in *An. funestus* (N = 92).

Only *An. funestus* and *An. gambiae* were considered in the microarray experiment because *An. arabiensis* is a minor contributor to malaria transmission in the study area [35, 36]. Transcriptomic analysis of *An. funestus* from the village of Kabirizi, revealed 789 probes (out of approximately 60,000) as significant using stringent criteria, of which 375 were over-expressed relative to the FANG susceptible strain (Fig 4A; S3 Table). Of these, 48 could be identified as

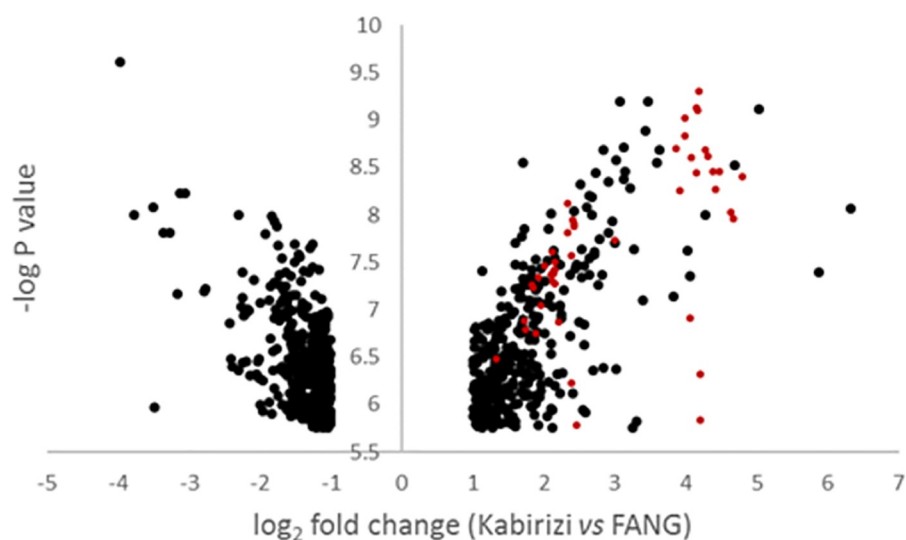

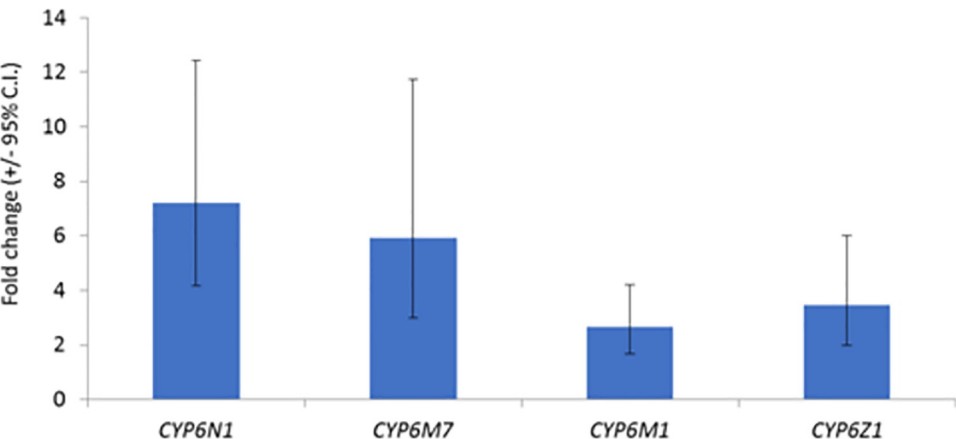

**Fig 4. Genes significantly over expressed in *An.funestus* from Kabirizi vs FANG laboratory susceptible colony.** (a) Volcano plot of all probes significant in experimental comparison (b) Relative expression levels of four candidate genes in qPCR. Red circles show over expressed gene probes targeting P450 genes.

having a possible role in detoxification processes, with 47 of the probes coming from just five P450 genes (each represented by multiple probes).

Two P450s, CYP6N1 and CYP6M7, were among the most statistically significant and highly expressed of all genes; expressed at levels 17 and 21-fold higher, respectively, than the FANG strain. The other significant genes represented by multiple significant probes were CYP6Z1 and CYP6M1 (each 4 to 5 times higher than FANG; S4 Table). Each of the genes was confirmed as being significantly over expressed by qPCR (S5 Table), and whilst fold-change values were lower than those in the microarray, CYP6N1 and CYP6M7 remained as the more highly expressed (Fig 4B).

In transcriptome analysis of *An. gambiae* from Kakoma village, 562 probes were consistently significantly differentially expressed in relation to both of the susceptible strains included (Fig 5A; S6 Table). Of these, 87 (65 of which were over expressed) were identifiable as possible detoxification-related genes, including members of the three major metabolic gene subfamilies P450s, GSTs and COEs, as well as transporter genes and alcohol dehydrogenases.

From these significant detoxification genes, P450s were the most common class, with CYP6M2, CYP6Z3, CYP6P3, CYP6P4, CYP6AA1 and CYP9K1 being the most notable (expression range between 4 and 11 higher than the susceptible strains). These P450 genes, along with VATPase—the most over expressed gene ($\approx$88-fold) and also two highly under-expressed detoxification genes GSTE2 and CYP9J5 were chosen for qPCR analysis, along with an additional P450, CYP6M1, that was not significantly over expressed and which was included as a negative control (S7 Table). Excluding the VATPase gene, which showed more than 20-fold lower expression in qPCR than the microarray results, there was a broad agreement between datasets (Pearson's r = 0.58, N = 18). Most genes were expressed at similar or greater levels in qPCR than in the microarray (Fig 5A).

Moreover, all genes, barring the negative control CYP6M1, were significantly over-expressed in comparison with either of the colonies (S8 Table). The physically-neighboring genes CYP6P3 and CYP6P4 were most strongly and consistently over expressed, followed by CYP6M2 and CYP9K1 (Fig 5B). Each of these *An. gambiae* P450 genes are known to metabolize pyrethroids.

## Discussion

At both baseline and after two years of interventions, *An. gambiae* showed an extremely high frequency of resistance to both permethrin and lambda-cyhalothrin, with only one instance of permethrin mortality being greater than 10%. In 2017, *An. funestus* were also tested and showed moderate resistance level (31% mortality). Based on resistance ratios, both *An. gambiae s.s*, *An. arabiensis* and *An. funestus* were highly resistant to insecticides tested.

WHO recommends PBO LLIN deployment in areas of intermediate resistance (10–80% mortality). The strength of resistance is not considered by the WHO recommendation because data were unavailable to populate the malaria model predictions on which the recommendation was based [37]. However, strength of resistance is potentially a more crucial metric because vectors that might just survive a diagnostic dose (but not a dose of greater concentration) are formally classed as resistant. Slight increases in dosage might kill them and they may represent little threat to the operational effectiveness of ITNs [38, 39]. Although resistance intensities may be estimated in different ways, in Muleba *An. gambiae* and *An. funestus* showed estimated resistance ratios (LC$_{50}$ dose points) of 63 and 39 respectively, similar to those reported for a resistant population from Uganda [38]. Although we did not use the practical method of 5× and 10× diagnostic concentrations for assessing resistance intensity, we were able to observe mortality rates at different doses of permethrin. *An. funestus* showed

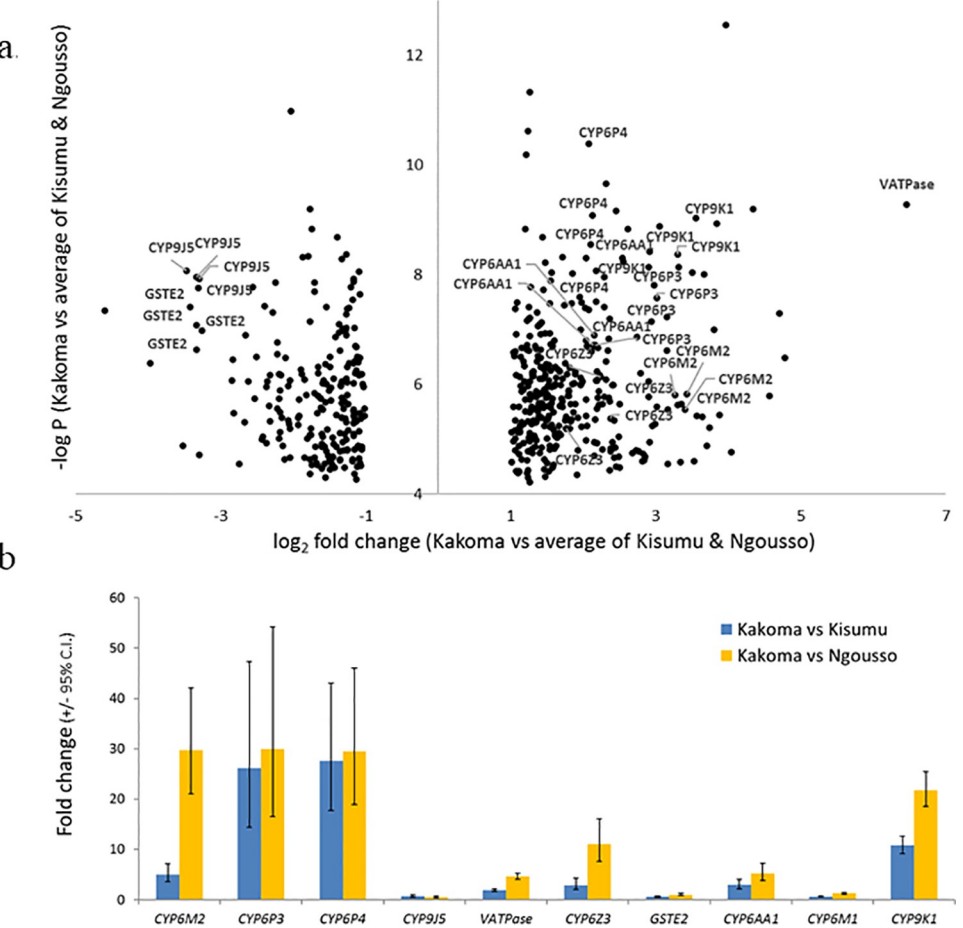

**Fig 5. Genes significantly over expressed in *An. gambiae* from Kakoma vs Kisumu and Ngousso laboratory susceptible colonies.** (a) Volcano plot of all probes significant in all experimental comparisons (average of the two experiments). (b) Relative expression levels of candidate genes in qPCR. Note that CYP6M2 and CYP6Z2 are significant vs Kisumu despite lower fold changes than vs Ngousso.

suspected resistance when the concentration was 10 times the diagnostic dose and would be interpreted as indicative of high resistance intensity [25]. For *An. gambiae* 20% survived the concentration that was slightly greater than 10x and 3% survived the concentration which was 40x the diagnostic dose.

With such a profile mosquitoes would be able to successfully feed through standard pyrethroid LLINs [40] which may explain the small impact of these nets in the Tanzania trial [15]. The addition of PBO in synergist assay resulted in a partial restoration of susceptibility to permethrin in both *An. gambiae* and *An. funestus* which indicates the involvement of P450. This was confirmed by the identification of multiple P450 genes over expressed in *An. gambiae*. CYP6M2, CYP6P3, CYP6P4, CYP9K1 are particularly notable because they have been frequently associated with resistance in transcriptomic studies [41, 42] and demonstrated to metabolize pyrethroids [43, 44]. Evidence that these genes appear key to resistance in *An. gambiae* has come from West and West Central Africa [11]. The P450s identified in these studies were also the key ones in Muleba, suggesting either convergent evolution or spread of mechanisms between both sides of the continent.

The most significantly over expressed genes in *An. funestus* were dominated by CYP6 sub-family P450s, most notably CYP6N1, CYP6M7, CYP6M1 and CYP6Z1, all of which have been previously associated with pyrethroid resistance [45, 46]. The two best-known pyrethroid-associated P450s in *An. funestus*, CYP6P9a and CYP6P9b [47] were, however, not over expressed in Muleba, which thus appears to be a population in which CYP6M7 acts in their stead and which metabolizes pyrethroids with equally high efficiency [14]. The gene expression data for the two important Muleba malaria vectors thus provides strong evidence to validate the third WHO criterion, for deployment.

Interestingly, carbamate resistance increased significantly in An. gambiae between 2014 and 2017, despite cessation of carbamate use for control prior to the baseline collections.

The presence of acetylcholinesterase (*Ace1*) target site mutation has been involved in resistance to organophosphates and/or carbamates in Anopheles populations [48]. Bendiocarb resistance was already observed in the study site in 2011 while *Ace1* mutations was not detected at this time in local *An. gambiae* [18]. In the present study, *Ace 1* was not investigated and could have contributed to the change in bendiocarb resistance. Pyrethroid-driven over expression of P450s may cause or perhaps combine with *Ace 1* mutations to produce bendiocarb resistance, with the primary cross-resistance candidate genes CYP6M2 and CYP6P3 in *An. gambiae's* sister species *An. coluzzii* [48], and CYP6Z1 in *An. funestus* [45] all significantly over expressed in Muleba populations.

In Muleba the mechanistic profile of both *An. gambiae* and, especially, *An. funestus*, which lacks pyrethroid target site mutations, appears dominated by over expression of key candidate P450s. A crucial point is that genes providing evidence for a P450-based resistance mechanism are not just those shown to be over expressed in the current or even previous studies but have been subject to functional validation to demonstrate their role in pyrethroid detoxification and/or resistance, which is the case for genes in both focal species of this study. Also, in this study, the frequency of pyrethroid resistance in Muleba was much greater than 90% in *An. gambiae* and in the cluster randomised trial PBO-LLINs were significantly more effective than standard LLINs [15]. PBO-LLINs could, therefore, still be recommended where resistance is greater than 80% and metabolic resistance is prevalent.

## Conclusions

*A posteriori* resistance profiling shows that malaria vector populations in the study area had high resistance intensities yet malaria prevalence in the PBO-LLIN was reduced by 44% compared to standard LLIN [15]. Revision of the condition specified by the WHO for the deployment of PBO-LLIN should be considered as evidence on the efficacy of these nets accumulates.

## Supporting information

**S1 Fig. Testing timeline.**
(TIF)

**S2 Fig. Interwoven microarray experimental loop design.**
(TIF)

**S1 Table. Primers used in *An. funestus* quantitative real-time PCR.**
(DOC)

**S2 Table. Primers used in *An. gambiae* quantitative real-time PCR.**
(DOC)

**S3 Table. Full results from *An. funestus* microarray analyses.**
(XLSX)

**S4 Table. Characteristics of genes from microarrays analysis that were carried forward for *An. funestus* qPCR.**
(DOC)

**S5 Table. Quantitative PCR results for *An. funestus*.**
(XLSX)

**S6 Table. Full results from *An. gambiae* microarray analyses.**
(XLSX)

**S7 Table. Characteristics of genes from microarrays analysis that were carried forward for *An. gambiae* qPCR.**
(DOC)

**S8 Table. Quantitative PCR results for *An. gambiae*.**
(XLSX)

## Acknowledgments

We are greatly indebted to the PAMVERC field workers in Muleba for their active participation in mosquito collections, testing and processing. We express our sincere thanks to PAMVERC insectary and molecular laboratory staff in Moshi for rearing mosquitoes, processing and preserving mosquitoes in RNALater. The assistance from the community leaders and households from which mosquitoes were collected is highly appreciated. Last but not least, a lot of thanks are to Mr. Rabieth Shani and Mrs. Fridah Temba, the PAMVERC administrators in Muleba and Moshi respectively for settling logistics issues throughout the study.

## Author Contributions

**Conceptualization:** Johnson Matowo, Natacha Protopopoff.

**Formal analysis:** Johnson Matowo, David Weetman, Natacha Protopopoff.

**Funding acquisition:** Johnson Matowo, Natacha Protopopoff.

**Investigation:** Johnson Matowo, David Weetman, Patricia Pignatelli, Alexandra Wright, Franklin Mosha, Mark Rowland.

**Methodology:** Johnson Matowo.

**Supervision:** Johnson Matowo.

**Writing – original draft:** Johnson Matowo.

**Writing – review & editing:** David Weetman, Jacques D. Charlwood, Robert Kaaya, Boniface Shirima, Oliva Moshi, Eliud Lukole, Jacklin Mosha, Alphaxard Manjurano, Franklin Mosha, Mark Rowland, Natacha Protopopoff.

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
