## [Decision Letter · Decision Letter 0]

26 Apr 2021

PONE-D-21-08461

Insecticide resistance characteristic of Anopheles vector species successfully controlled by deployment of pyrethroid and PBO long lasting insecticidal treated nets (LLINs) in Tanzania

PLOS ONE

Dear Dr. Johnson Matowo,

Thank you for submitting your manuscript to PLOS ONE. After careful consideration, we feel that it has merit but does not fully meet PLOS ONE’s publication criteria as it currently stands. Therefore, we invite you to submit a revised version of the manuscript that addresses the points raised during the review process.

Please clarify text, to clearly indicate mosquito species in each session

Please discuss the possible relevance of a number of additional insecticide resistance molecular markers not accessed in the study  

We look forward to receiving your revised manuscript.

Kind regards,

John Vontas

Academic Editor

PLOS ONE

Journal Requirements:

3)  Please clarify whether any permits were necessary/obtained to collect mosquitoes from the specified areas.

4) Please provide a citation for the clinical trail that mosquitoes were collected during.

5) We note that you have stated that you will provide repository information for your data at acceptance. Should your manuscript be accepted for publication, we will hold it until you provide the relevant accession numbers or DOIs necessary to access your data. If you wish to make changes to your Data Availability statement, please describe these changes in your cover letter and we will update your Data Availability statement to reflect the information you provide.

6) Please amend either the title on the online submission form (via Edit Submission) or the title in the manuscript so that they are identical.

7) Please upload a copy of Supporting Information Figure S1 which you refer to in your text on page 25.

Reviewers' comments:

Reviewer's Responses to Questions

**Comments to the Author**

1. Is the manuscript technically sound, and do the data support the conclusions?

Reviewer #1: Yes

Reviewer #2: Yes

2. Has the statistical analysis been performed appropriately and rigorously? 

Reviewer #1: Yes

Reviewer #2: Yes

3. Have the authors made all data underlying the findings in their manuscript fully available?

Reviewer #1: Yes

Reviewer #2: Yes

4. Is the manuscript presented in an intelligible fashion and written in standard English?

Reviewer #1: Yes

Reviewer #2: Yes

5. Review Comments to the Author

Reviewer #1: The research article by Matowo et al is a comprehensive study regarding the insecticide status and related mechanisms operating in a targeted area in Tanzania, where a permethrin/PBO co treated LLINs trial occured. It provides important and operationally relevant information regarding insecticide resistance status which are applicable to relevant trials and field settings. The manuscript is well written and well organized and I only have the following suggestions to make:

1. The relevance OPs/ carbamates bioassays performed in this study (otherwise focusing on pyrethroid resistance) could be more elaborate to introduce the reader to the concept of cross resistance. Since no molecular diagnostics for OPs/carabamates have been performed, target site mechanisms should be discussed as a possible mechanism. The authors mention that no ace mutations had been detected in the area based on data from 2013, but this could have changed in 5-10 years’ time.

2. PBO experiments show clear metabolic resistance, but this could be, as in many other cases, a synergistic phenomenon operating in parallel with other resistance mechanisms (e.g. target site mutations like N1575Y and cuticle resistance).

3. In some instances, it is difficult to distinguish to which mosquito species the authors refer to (An. gambiae ss, An. coluzzii, or hybrid forms).

4. It would be relevant to discuss why the newly defined CYP6P9a/9b SNPs associated with resistance in An. funestus were not assessed (i.e., no CYP6P9a overexpression detected).

5. The continued dependence on pyrethroid-treated LLINs is very relevant and could be discussed in more details (logistics: limited supplies of co-PBO treated nets, high cost, distribution issues, other factors?)

6. A supplementary Table with the list of primers used in this study would be useful.

Reviewer #2: The paper entitled "Insecticide resistance characteristic of Anopheles vector species successfully controlled by deployment of pyrethroid and PBO long lasting insecticidal treated nets in Tanzania" is an interesting piece of work from experts in the field of insecticide resistance and vector control. Cluster randomized trial demonstrated that permethrin LLINs co-treated with the mixed function oxidase inhibitor (PBO) was more efficient than the net treated only with pyrethroid. Bioassay data and molecular analysis undertaken in this study to determine the underlying mechanism of pyrethroid resistance in Anopheles gambiae and An. funestus. Both data were in agreement with LLINs efficacy data. The manuscript well written and the data nicely presented and support the final conclusion. I have no hesitation to recommend this work in its form for publication in PLOS ONE.

6. PLOS authors have the option to publish the peer review history of their article (what does this mean?). If published, this will include your full peer review and any attached files.

Reviewer #1: No

Reviewer #2: No

---

## [Author Response · Author response to Decision Letter 0]

1 Jul 2021

All specific reviewer and editor comments have been addressed accordingly in a letter on Response to reviewers 

(attached)

---

## [Decision Letter · Decision Letter 1]

5 Oct 2021

PONE-D-21-08461R1Insecticide resistance characteristic of Anopheles vector species successfully controlled by deployment of pyrethroid and PBO long lasting insecticidal treated nets in TanzaniaPLOS ONE

Dear Dr. Matowo,

Thank you for submitting your manuscript to PLOS ONE. After careful consideration, we feel that it has merit but does not fully meet PLOS ONE’s publication criteria as it currently stands. Therefore, we invite you to submit a revised version of the manuscript that addresses the points raised during the review process.

We look forward to receiving your revised manuscript.

Kind regards,

Nicholas C. Manoukis

Academic Editor

PLOS ONE

Journal Requirements:

Additional Editor Comments (if provided):

Thank you for your resubmission. I was asked to take over as AE, and happy to do so. Please note carefully the comments from reviewer #3- the relation between the results in the paper and ultimate real world effectiveness of LLINs seems quite indirect - here the actual effectiveness is not measured in the field, rather resistance of mosquitoes from field populations is measured and quantified. To illustrate what I mean, note the sentence from the abstract starting on line 19:

"A cluster randomized trial in Muleba district, Tanzania demonstrated that permethrin LLINs co-treated with piperonyl butoxide (PBO), a synergist that can block pyrethroid-metabolizing enzymes in the mosquito, had much greater efficacy than pyrethroid-only nets."

This sentence implies more that what was measured in this study. Rather, the data show that "permethrin LLINs co-treated with piperonyl butoxide (PBO), a synergist that can block pyrethroid-metabolizing enzymes in the mosquito, was more effective in knocking down and killing field-derived populations of An. gambiae and An. funestus in laboratory assays than pyrethroid-only nets".

The current title ("Insecticide resistance characteristic of Anopheles vector species successfully controlled by deployment of pyrethroid and PBO long lasting insecticidal treated nets in Tanzania") is likewise misleading, reflecting some of the same issues- particularly the words "deployment", "control", and "in Tanaznia", all of which suggest overall effectiveness of an intervention in the field. As far as I can tell from the MS, nothing was deployed in the field and no control of malaria vectors in nature was measured. A more accurate title is needed, focusing on what was done: Increased mortality of field-derived populations of Anopheles mosquitoes from Tanzania when exposed to pyrethroid and PBO long lasting insecticidal nets.

I do not understand how the conclusion section follows from the results presented. Malaria prevalence was not measured in this study as far as I can tell.

To be absolutely clear as I can be, the issue comes down to this: The overall effectiveness of a LLIN is measured in the field- many factors besides the ability of resistant mosquitoes to evade death from the insecticide can affect effectiveness (e.g. frequency of resistant genotypes, relative use rates of treated bednets, predator assemblages, sub-lethal effects, etc), and many of these are not assessed in this study.

None of this is to say the study is not worth publishing. What you have submitted is research on the fine-scale mechanisms of insecticide resistance in some malaria vector populations in Tanzania- results that have relevance to treated bed net effectiveness, but are not the whole story. Please adjust the focus of your presentation accordingly to increase the clarify on what was actually done and its relation to the possible situation in the field.

Reviewers' comments:

Reviewer's Responses to Questions

**Comments to the Author**

1. If the authors have adequately addressed your comments raised in a previous round of review and you feel that this manuscript is now acceptable for publication, you may indicate that here to bypass the “Comments to the Author” section, enter your conflict of interest statement in the “Confidential to Editor” section, and submit your "Accept" recommendation.

Reviewer #1: All comments have been addressed

Reviewer #2: All comments have been addressed

Reviewer #3: (No Response)

2. Is the manuscript technically sound, and do the data support the conclusions?

Reviewer #1: Yes

Reviewer #2: Yes

Reviewer #3: Partly

3. Has the statistical analysis been performed appropriately and rigorously? 

Reviewer #1: Yes

Reviewer #2: Yes

Reviewer #3: No

4. Have the authors made all data underlying the findings in their manuscript fully available?

Reviewer #1: Yes

Reviewer #2: Yes

Reviewer #3: (No Response)

5. Is the manuscript presented in an intelligible fashion and written in standard English?

Reviewer #1: Yes

Reviewer #2: Yes

Reviewer #3: Yes

6. Review Comments to the Author

Reviewer #1: (No Response)

Reviewer #2: The authors have responded to reviewers comments adequately. Since I have no further comments, I have no hesitation to recommend this work for publication.

Reviewer #3: Matowo et al. examined the incidence, extent, and management of pyrethroid resistance in Anopheles mosquitoes during 2014-2016 in five villages of the Muleba district in North Tanzania. As pyrethrioid resistance to permethrin treated insecticidal nets has reduced the efficacy of this approach, augmentation using piperonyl butoxide, a synergist that blocks pyrethroid metabolizing enzymatic activity, was investigated. Four villages were selected to receive 1 of 4 treatments: 1) long-lasting insecticide (permethrin) treated netting (LLIN), 2) LLIN + piperonyl butoxide (Py-PBO), 3) LLIN + and indoor residual spray (IRS) using an organophosphate product Actellic, 4) LLIN + Py PBO + IRS. They then conducted sampling to determine if the nets differentially affected mosquito populations in each village. Using flies from the collection the authors conducted a series of tests: a) a resistance assay using single diagnostic concentrations of permethrin, lambda-cyhalothrin, bendiocarb, and pirimiphos-methyl; b) a synergist bioassay aimed at determining how pre-exposure to piperonyl butoxide affected permethrin resistance values, c) a resistance intensity dose response assay aimed at establishing resistance curves for two mosquito species from two villages both treated with the permethrin netting LLIN treatment alone, Kakoma and Kabirizi. They also conducted molecular and genetic analyses of wild collected An. gambiae and An. Funestus to determine d) genotyping for kdr and GSTe2 mutations; e) transcriptome analysis of pyrethroid resistant mosquitoes; and f) candidate gene expression analysis.

Generally, the manuscript is well written and easy to read. The introduction fully introduces the concepts needed to understand the goals of the project and the methods are well written as far as what was provided. However, there are a number of problems with the manuscript that make evaluation of the claims difficult. Most importantly, while the focus of the paper is the efficacy of different netting treatments, the data on trapping efficacy is entirely missing. Even after review, it is unclear to me how the authors planned to assess treatment efficacy and there no data provided to show that the netting was effective at management mosquito populations at all. Rather, the authors devote the results to lab assessments of resistance values using flies caught from these villages. However, even with those data, sampling dates are not provided, and statistical comparisons between groups for the data that are presented are absent. Indeed, while there is a suppression of susceptibility to pyrethroids, in comparison to other chemical classes, between sites, the resistance values appear similar and without statistical comparison, there is no way to determine whether differential treatments affected resistance. In fact, the resistance of flies from the single LLIN treatment at Kakoma appeared similar to those that did include the PBO treatment at Kishuro and Kiteme. The authors could either, A) provide those missing data, if they are available and introduce new analyses, or B) the authors may want to change the wording of their claims and make their manuscript more focused on their explicit goals.

Please see my more detailed line comments below:

Ln 68: Generally, readers struggle to keep multiple acronyms straight. I would recommend writing out the insecticide treatments instead. It would really make the manuscript more accessible. However, if you feel that the acronyms are necessary, please consider simplifying them. Can you simplify Py-PBO to just PBO? Having such long acronyms makes deciphering the treatments tiring.

Ln 70: Please make sure to define IRS, Indoor residual spray, for the reader. I would also elaborate on this approach too. Why is it used? How does it compare to netting? Does combining approaches and insecticide classes affect resistance and resistance management?

I would also reiterate here that the pyrethroid used in the netting was permethrin.

Lns 91-94: At this stage it would be helpful describe in detail what you hypotheses were. What did you expect to learn from the deployment of the different treatments to the 5 villages in Muleba? Summarize in a few sentences what you did and why.

Ln 96: Why is this first section of the methods in quotes? Please remove. I would also give this section a heading of its own “Consent and ethical clearance” or something to that effect.

Ln 104-117: There is a lot of information missing from this section. Please include 1) the dates that each treatment were deployed, 2) how long they were deployed, 3) whether and when the treatments were reapplied or serviced, 4) the names of the pesticides used, 5) the manufacturer information of those pesticide products, 6) the netting information (mesh gauge, brand), 7) the concentrations and volume applied both to the netting and the indoor sprays.

Figure 1: Can you include the sampling location data on the map? How many sample sites were in each village? Just one or multiple?

Ln 119: Please reconsider this section organization. I think there this too much here and some of the information belongs in a different section altogether. Please only include sample collection and id info here. Move all the information in the middle paragraph to a new section on lab susceptible strains.

Ln 121: What how many samples were collected? For the data to be robust (fig2) there should be multiple collections from multiple sites in each village. Was this done? If so, please describe.

Ln 127 and going forward: Please refrain from referring to these strains by their village names. It makes keeping them straight nearly impossible for the reader. Instead, please refer to them as the susceptible strain for each species.

Ln 141: For each of these sections going forward I would consider giving a brief 1-2 sentence description of why these assessments were conducted. This helps the reader understand why you did what you did and better interpret the results later on. Including a clear description of your hypotheses in the introduction will help with this too.

Ln 144-145: Because you refer to the class of insecticides throughout the article, it would be helpful at this stage to include those class names here with each of the insecticides you tested. For instance, you could include in parentheses (pyrethroid) after permethrin, or (carbamate) after bendiocarb.

You also need to include the source of the chemicals you used here. Where did you purchase them? Who was the manufacturer? This is important for replication.

What are these concentrations in ug/ml values? This will help these data make sense given the concentration values reported later on in fig. 3.

I agree with one of the other reviewers too, that including information on why these insecticides were selected for analysis is important. Right now that information is absent. Is bendiocarb at risk for cross resistance? Are lambda-cyhalothrin and bendiocarb used in mosquito management and if so how? Water treatments? Fogging? Netting?

Ln 152: Why were PBO and permethrin administer separately rather than concomitantly? This method is not explained.

Ln 166: Please be sure to fully explain what the goals of each genetic analysis were. In other words, why do we care about these mutations?

Also please include the collection dates and location for the insects used in these analyses.

Ln 182: Same as the last comment. Please describe why you conducted this transcriptome analysis, what you attempted to learn, and why that is important.

Also please include collection dates and location.

Ln 246: Is there no trapping data you can include here? How many flies were recovered from each village during each sample? How did you evaluate the efficacy of each treatment?

Ln 251: Please refer to the treatments instead of or alongside the village names.

Ln 253: Please report how many flies were collected at each location, if possible.

Lns 257-265: These is a lot of information here. I could consider making a table instead and show the Lc50 values of each species at each location for each year.

Ln 277: What was the diagnostic dose? Providing this information is important for comparison.

Ln 289: Please consider adding an additional footnote to table 1. Please include how RR is calculated. I would also consider adding information to the methods and materials describing the significance of different RR values. The WHO describes that values less than 5 indicated susceptibility while values greater than 5 indicate resistance in Aedes mosquitoes. Is this the same for Anopheles?

Fig. 2. Please add statistical analysis between groups. Logistic regression and posthoc analysis among sites for each pesticide would help determine if these differences in mortality were in fact different.

I would also add the species label to the y-axis for clarity, and change the village names to treatment labels

Fig. 3. I would again, go back and use logistic regression to compare survival and report he LC50 values here on the figure. For good measure, you could also denote the diagnostic dose here on the figure for each species.

Fig 4. In figure captions, please refer to treatment at the Kabirizi village. Also include what year those samples were collected.

Can you conduct a comparison of the CT values for Kabarizi vs the lab susceptible population (FANG)? A t-test for each gene may suffice. Otherwise, we don’t know if these differences are statistically significant or not.

Fig. 5. What year were these samples collected? Refer to the lab susceptible populations rather than village names (Kisumu or Ngousso). Again, run a stats comparison for the village vs lab susceptible. Also, the DPI resolution of this figures is a little low, making the figure labels difficult to read, but that could just be on my end.

Ln 358: This sentence appears to have a typo. Change to “ The addition of PBO in the synergist assay…” ?

Ln 380: The authors state that the mechanism of resistance appears the same for the pyrethroids and carbamate because bendiocarb resistance was present in An. funestus. However, this may not be the case and the experiments reported here to do not evaluate this hypothesis. Be careful not to overextend on your claims.

Ln 389: Where are these data reported? Where does this 90% values come from?

7. PLOS authors have the option to publish the peer review history of their article (what does this mean?). If published, this will include your full peer review and any attached files.

Reviewer #1: No

Reviewer #2: No

Reviewer #3: No

---

## [Author Response · Author response to Decision Letter 1]

24 Nov 2021

All comments from the editor and reviewer(s) have been well addressed

---

## [Decision Letter · Decision Letter 2]

14 Dec 2021

PONE-D-21-08461R2Expression of pyrethroid metabolizing P450 enzymes characterizes highly resistant Anopheles vector species targeted by successful deployment of PBO-treated bednets in TanzaniaPLOS ONE

Dear Dr. Matowo,

Thank you for submitting your manuscript to PLOS ONE. After careful consideration, we feel that it has merit but does not fully meet PLOS ONE’s publication criteria as it currently stands. Therefore, we invite you to submit a revised version of the manuscript that addresses the points raised during the review process.

We look forward to receiving your revised manuscript.

Kind regards,

Nicholas C. Manoukis

Academic Editor

PLOS ONE

Journal Requirements:

Additional Editor Comments (if provided):

Nice job addressing reviewer comments. Please address the last remaining issues with special attention to including stats and enough information on other variables already published to make the paper a stand-alone product. I look forward to seeing another revision.

Reviewers' comments:

Reviewer's Responses to Questions

**Comments to the Author**

1. If the authors have adequately addressed your comments raised in a previous round of review and you feel that this manuscript is now acceptable for publication, you may indicate that here to bypass the “Comments to the Author” section, enter your conflict of interest statement in the “Confidential to Editor” section, and submit your "Accept" recommendation.

Reviewer #3: (No Response)

2. Is the manuscript technically sound, and do the data support the conclusions?

Reviewer #3: Yes

3. Has the statistical analysis been performed appropriately and rigorously? 

Reviewer #3: Yes

4. Have the authors made all data underlying the findings in their manuscript fully available?

Reviewer #3: Yes

5. Is the manuscript presented in an intelligible fashion and written in standard English?

Reviewer #3: Yes

6. Review Comments to the Author

Reviewer #3: Matowo et al. has addressed most of the comments I made in the previous review round. The new title is good and is more descriptive of their current project. Adding a few sentences here and there in the methods to keep the reader focused on what they did and why is also a big improvement. It reads much better now. Good job with that. Just a few comments remain.

First, in the abstract, I would consider reworking line 32. The current manuscript reads “…had much greater efficacy than pyrethrioid-only nets.” This still sounds like you tested the efficacy of pesticides on reducing mosquito populations rather than defining resistance characteristics. Be more specific with what you are referring to when you say “ efficacy.” Efficacy of what? Please consider changing to something like this, to avoid confusion for the reader: “nets treated with PBO reduced the incidence of or severity of resistance more than pyrethroid only nets” or something to that effect.

Ln 93: removed comma after the word study

Lns 102-107: Please remove those quotation marks

Ln 110-124: While I appreciate that you already included the pesticide trial information in a previously published paper, you still need to include this information again as this is a separate paper. Please include the names of the pesticides and the rates they were applied. For more information related the trial, please explicitly refer the reader to Protopopoff 2018. I would also consider giving a brief 1-2 sentence summary of that project so the reader can follow how these are connected. It’s best not to assume the reader will be familiar with your other work. The current paper should be self-explanatory. The reader shouldn’t have to be familiar with your other projects to understand what you did here and at the very least, should some information be severely redundant to explain, them clearly indicate where the reader should go for that information.

Ln 140: Good job breaking out these sections in the methods. Reads much clearer now.

Ln 150: I really like this supplemental figure!

Ln 244-267: Maybe I am missing it, but in the rebuttal, the authors state that they used t-tests to analyze differences in CT values between groups and they refer to the stats section for that information. I still don’t see that here. Please include.

Ln 332 & ln 358: In both places the authors refer to significant differences. I assume this refers to the t-tests they ran? Where are the values for those stats? Please include.

7. PLOS authors have the option to publish the peer review history of their article (what does this mean?). If published, this will include your full peer review and any attached files.

Reviewer #3: No

---

## [Author Response · Author response to Decision Letter 2]

17 Dec 2021

All issues that were raised by Editor and Reviewer #03 have been adequately addressed

---

## [Decision Letter · Decision Letter 3]

21 Dec 2021

Expression of pyrethroid metabolizing P450 enzymes characterizes highly resistant Anopheles vector species targeted by successful deployment of PBO-treated bednets in Tanzania

PONE-D-21-08461R3

Dear Dr. Matowo,

We’re pleased to inform you that your manuscript has been judged scientifically suitable for publication and will be formally accepted for publication once it meets all outstanding technical requirements.

Kind regards,

Nicholas C. Manoukis

Academic Editor

PLOS ONE

Additional Editor Comments (optional):

Great job, and congratulations on a nice study!

Reviewers' comments:

Reviewer's Responses to Questions

**Comments to the Author**

1. If the authors have adequately addressed your comments raised in a previous round of review and you feel that this manuscript is now acceptable for publication, you may indicate that here to bypass the “Comments to the Author” section, enter your conflict of interest statement in the “Confidential to Editor” section, and submit your "Accept" recommendation.

Reviewer #3: All comments have been addressed

2. Is the manuscript technically sound, and do the data support the conclusions?

Reviewer #3: Yes

3. Has the statistical analysis been performed appropriately and rigorously? 

Reviewer #3: Yes

4. Have the authors made all data underlying the findings in their manuscript fully available?

Reviewer #3: Yes

5. Is the manuscript presented in an intelligible fashion and written in standard English?

Reviewer #3: Yes

6. Review Comments to the Author

Reviewer #3: Great job on the revisions! This is a fine paper and will be a meaningful contribution to the field. Best of luck in the future and happy holidays.

7. PLOS authors have the option to publish the peer review history of their article (what does this mean?). If published, this will include your full peer review and any attached files.

Reviewer #3: No

---

## [Editor Report · Acceptance letter]

13 Jan 2022

PONE-D-21-08461R3 

Expression of pyrethroid metabolizing P450 enzymes characterizes highly resistant *Anopheles* vector species targeted by successful deployment of PBO-treated bednets in Tanzania

Dear Dr. Matowo:

I'm pleased to inform you that your manuscript has been deemed suitable for publication in PLOS ONE. Congratulations! Your manuscript is now with our production department. 

Kind regards, 

on behalf of

Dr. Nicholas C. Manoukis 

Academic Editor

PLOS ONE